# Baseline CTC Count as a Predictor of Long-Term Outcomes in High-Risk Prostate Cancer

**DOI:** 10.3390/jpm13040608

**Published:** 2023-03-30

**Authors:** Wojciech A. Cieślikowski, Piotr Milecki, Monika Świerczewska, Agnieszka Ida, Michał Kasperczak, Agnieszka Jankowiak, Michał Nowicki, Klaus Pantel, Catherine Alix-Panabières, Maciej Zabel, Andrzej Antczak, Joanna Budna-Tukan

**Affiliations:** 1Department of Urology, Poznan University of Medical Sciences, 62-385 Poznan, Poland; 2Department of Electroradiology, Poznan University of Medical Sciences, 61-868 Poznan, Poland; 3Department of Histology and Embryology, Poznan University of Medical Sciences, 60-781 Poznan, Poland; 4Department of Tumor Biology, University Medical Centre Hamburg-Eppendorf, 20246 Hamburg, Germany; 5Laboratory of Rare Human Circulating Cells, University Medical Center, 34093 Montpellier, France; 6Division of Histology and Embryology, Department of Human Morphology and Embryology, Wroclaw Medical University, 50-368 Wroclaw, Poland; 7Division of Anatomy and Histology, University of Zielona Góra, 65-046 Zielona Góra, Poland

**Keywords:** prostate cancer, circulating tumor cells, prognosis, overall survival, metastasis-free survival

## Abstract

The aim of the present study was to verify whether the baseline circulating tumor cell (CTC) count might serve as a predictor of overall survival (OS) and metastasis-free survival (MFS) in patients with high-risk prostate cancer (PCa) during a follow-up period of at least 5 years. CTCs were enumerated using three different assay formats in 104 patients: the CellSearch^®^ system, EPISPOT assay and GILUPI CellCollector. A total of 57 (55%) patients survived until the end of the follow-up period, with a 5 year OS of 66% (95% CI: 56–74%). The analysis of univariate Cox proportional hazard models identified a baseline CTC count ≥ 1, which was determined with the CellSearch^®^ system, a Gleason sum ≥ 8, cT ≥ 2c and metastases at initial diagnosis as significant predictors of a worse OS in the entire cohort. The CTC count ≥ 1 was also the only significant predictor of a worse OS in a subset of 85 patients who presented with localized PCa at the baseline. The baseline CTC number did not affect the MFS. In conclusion, the baseline CTC count can be considered a determinant of survival in high-risk PCa and also in patients with a localized disease. However, determining the prognostic value of the CTC count in patients with localized PCa would optimally require longitudinal monitoring of this parameter.

## 1. Introduction

Prostate cancer (PCa) is the most common male malignancy, with the global number of new cases and deaths approximating 1.6 million and nearly 370 thousand annually, respectively [1,2,3,4].

Nowadays, the diagnosis of PCa is based on a histopathological examination of the biopsy specimen, and prognosis and treatment depend on the biopsy Gleason sum and serum concentration of prostate-specific antigen (PSA) [5]. While the PSA has been used for decades in diagnosing PCa and monitoring treatment outcomes, its clinical application is limited by its low specificity. Elevated levels of PSA are also reported in many non-malignant diseases of the prostate, such as benign hyperplasia, infections and prostatic infarction [6]. Additionally, PSA is encoded by an androgen-dependent gene; thus, any change in serum androgen level or direct modulation of the PSA gene by some therapeutic agents might lead to a false decrease in PSA concentration that is not associated with the natural history of PCa [7].

Circulating tumor cells (CTCs) are the cells that detach from the primary tumor mass or secondary metastatic tumor and reach peripheral circulation. As CTCs are vital for tumor spread, they are considered a highly attractive prognostic and predictive biomarker and a measure of treatment outcomes [8]. The detection of CTCs in peripheral blood, referred to as a “liquid biopsy”, constitutes a less invasive alternative to conventional tissue biopsy [9]. Considering a heterogeneous presentation, a considerable proportion of PCa being diagnosed at local stages but without the possibility of excluding systemic spread, and variable mechanisms of the latter [10], as well as the limitations of PSA mentioned above, CTC enumeration seems to be particularly useful in the monitoring of PCa [11].

Our previous study demonstrated that the CTC count accurately differentiated between patients with true localized and occult disseminated PCa at the time of initial diagnosis [12]. The aim of the present study, as an extension of the study mentioned above, was to verify whether the baseline CTC count might serve as a predictor of overall survival (OS) and metastasis-free survival (MFS) in the same group of patients during at least 5 years of follow-up.

## 2. Materials and Methods

### 2.1. Patients

The study patients were recruited in 2014–2016 at the Department of Urology, Poznan University of Medical Sciences, within the framework of the international multicenter project entitled “Circulating Tumor Cells as Biomarker for Minimal Residual Disease in Prostate Cancer” (CTC-SCAN). The aim of the CTC-SCAN project was to validate the number of CTCs isolated from a patient’s blood as a prognostic marker for relapse in high-risk PCa treated with primary radiotherapy. Only patients with newly diagnosed, non-metastatic PCa, representing the high-risk group according to the D’Amico criteria (cT ≥ 2c and/or PSA ≥ 20 ng/mL and/or their biopsy Gleason sum ≥ 8) [13], were eligible for the CTC-SCAN project. However, a considerable proportion of patients enrolled at our center presented with disseminated PCa at the time of diagnosis. This subgroup, along with the remaining participants with truly localized PCa, were qualified for the satellite study, the results of which have been published elsewhere [12].

### 2.2. Ethics

The protocol of the study was approved by the Local Bioethics Committee at the Poznan University of Medical Sciences (decision no. 28/13 of 3 January 2013), and written informed consent was sought from all the study subjects.

### 2.3. Diagnosis and Staging

A diagnosis of PCa was established based on taking patient history, physical examination, the measurement of serum PSA, and 10–12 core needle biopsies with the determination of a biopsy Gleason sum. To exclude soft tissue disease, abdominal and pelvic computed tomography scans were reviewed. Moreover, radionuclide bone scans were evaluated for the presence or absence of metastatic bone disease. The stage of the disease was defined according to the 7th edition of The American Joint Committee on Cancer (AJCC) staging manual [14].

### 2.4. CTC Enumeration

After establishing the diagnosis and before implementing any anti-cancer treatment, CTCs were enumerated using three different assay formats: the CellSearch^®^ system, EPISPOT assay and GILUPI CellCollector.

#### 2.4.1. CellSearch^®^ System

A 7.5 mL venous blood sample was collected from each patient in CellSave tubes (Janssen Diagnostics, Raritan, NJ, USA). CTCs were enumerated using the CellSearch^®^ Epithelial Cell Kit (Veridex, Warren, NJ, USA), with magnetic beads coated with antibodies against epithelial cell adhesion molecule (EpCAM) carried by a ferrofluid. Captured target cells were then immunostained with antibodies against cytokeratins (PanCK = CK8, 18, and 19) and the common leukocyte antigen CD45 to exclude leukocytes. Cells positive for EpCAM, cytokeratins with positive DAPI staining as a measure of nuclear integrity and those negative for CD45 were identified as CTCs.

#### 2.4.2. EPISPOT Assay

A 10 mL venous blood sample was obtained from each patient in an EDTA-coated tube. CTCs were enumerated and characterized using a dual fluorescent PSA/FGF2-EPISPOT assay. During the first step, CD45-positive cells were depleted from the sample using the RosetteSep system (StemCell Technology, Vancouver, BC, Canada). Subsequently, the CD45-depleted cell fraction was used for the proper EPISPOT assay. Briefly, the nitrocellulose membranes of the EPISPOT plates were coated with 1.04 µg/µL of anti-PSA H50 antibody (obtained from the Department of Biotechnology, University of Turku, Turku, Finland) and 0.5 µg/µL of anti-FGF2 500-M38 antibody (Peprotech, Cranbury, NJ, USA) diluted in PBS. Then, cells were seeded in each well and cultured for 48 h at 37 °C and 5% CO_2_. During this incubation step, the secreted marker proteins were directly captured on the antibody-coated membrane. Next, the cells were washed off, and the marker proteins were detected by secondary antibodies conjugated with fluorochrome dyes: 1.0 µg/µL of anti-PSA-H117-A555 antibody (obtained from the Department of Biotechnology, University of Turku, Finland) and 0.5 µg/µL of anti-FGF2 500-P18Bt labeled with biotin (Peprotech, Cranbury, NJ, USA) and subsequently with 1:20 anti-biotin–FITC antibody (Miltenyi Biotec, Bergisch Gladbach, Germany) diluted in 0.5% BSA/PBS. Immunospots were counted under a fluorescent microscope by video camera imaging and computer-assisted analysis (KS ELISPOT, Carl Zeiss Vision, Aalen, Germany): one immunospot corresponded to the fingerprint of one viable marker protein-secreting cell. For a positive control for PSA and FGF2 proteins, LNCap and NBTII cell lines were used, respectively (ATCC). All assays were conducted at the Department of Histology and Embryology, Poznan University of Medical Sciences, and the results were verified at the Laboratory of Rare Human Circulating Cells, University Medical Center in Montpellier (France).

#### 2.4.3. GILUPI CellCollector

A GILUPI CellCollector was inserted into the patient’s arm vein via a standard 20 gauge needle. During the 30 min application in the vein, up to 1500 mL of blood, including the respective CTCs, passed the 2 cm functionalized area of the GILUPI CellCollector. Passing CTCs were bound by the anti-EpCAM antibody, removed from the patient’s vein together with the GILUPI CellCollector and analyzed by immunostaining with fluorochrome-labeled anti-cytokeratin antibodies (both 1:50, anti-Pan-Cytokeratin A488, eBiosience; panCK-A488, Exbio, Prague, Czech Republic), anti-PSA antibody (1:80, anti-PSA-H117-A555 antibody; obtained from the Department of Biotechnology, University of Turku, Finland), an antibody against the leukocyte marker CD45 (1:25, CD45-A647, Exbio, Prague, Czech Republic), as well as staining of the nucleus (1 µg/mL, Hoechst 33258, Sigma, Burlington, MA, USA). The antibodies were conjugated with different fluorescent dyes, allowing for discrimination between the CTCs and leukocytes by fluorescence microscopy. CTCs were characterized as cytokeratin-positive, PSA-positive and CD45-negative nucleated cells.

### 2.5. Treatment and Follow-Up

The study patients were treated by radiotherapy at the Department of Radiotherapy, Greater Poland Cancer Center, Poznan, and followed up with according to current standards [15]. Except for those presenting with metastases at enrollment, all study patients received radiotherapy. All patients were treated with a linear accelerator using intensity-modulated radiation therapy/image-guided radiation therapy (IMRT-IGRT) technology combined with at least three months of androgen deprivation therapy (ADT) prior to external beam radiation therapy (EBRT), for a total time of ADT of 36 months. In the first phase, patients were irradiated using 2 Gy fractionation to the prostate gland, seminal vesicles and pelvic lymph nodes up to a total dose of 46 Gy, and then to the prostate and seminal vesicles for a total dose of 76–78 Gy. Lifelong ADT was provided to metastatic patients.

The minimum duration of the follow-up was five years, with the database lock in September 2022.

### 2.6. Statistical Analysis

The chi-square and the rank-sum tests were used to compare categorical and continuous variables, respectively. For the survival analysis, the Kaplan–Meier method was used to generate survival curves. The Cox proportional hazard regression method was used to fit univariate and multivariate survival models for the OS and MFS, the results of which are reported as hazard ratios (HR) with 95% confidence intervals (CI). Variables with significant *p*-values in the univariate analysis were then included in the multivariate analysis. All reported *p*-values are two-sided and were considered significant if less than 0.05. Calculations and graphics were obtained using STATA IC 16.1 (StataCorp, College Station, TX, USA).

## 3. Results

The initial analysis included the entire cohort of 104 PCa patients. The mean age of the group at the PCa diagnosis was 68.2 ± 6.5 years (range 51–86 years). The group included 19 (18%) patients diagnosed with a metastatic PCa; in the vast majority of these patients (*n* = 17), the metastases were found in the bones. The remaining 85 (82%) patients presented with localized disease at the time of diagnosis. The clinical characteristics of the study patients, both the entire cohort and the group with metastatic PCa, are presented in Table 1, in addition to the statistical characteristics of the CTC counts, which were determined with various techniques.

The study patients were followed up with for a median time of 78 months (range: 1–103 months).

During the follow-up period, newly developed metastases were found in 24 out of 85 (28%) patients initially diagnosed with localized PCa, with a 5 year MFS of 76% (95% CI: 66–84%; median MFS: not reached; lower quartile MFS: 63 months). The newly developed metastases were found primarily in the bones (*n* = 21), with isolated or concomitant soft tissue metastases present in seven patients.

No significant differences in clinical characteristics and CTC counts were found between patients who developed metastases during follow-up and those who did not except for a significantly higher proportion of tumors with a baseline cT ≥ 2c in the former group (Table 2).

The Kaplan–Meier curve for the MFS is shown in Figure 1a. The analysis of Cox proportional hazard models demonstrated that cT was the only significant predictor of MFS among baseline clinical characteristics (Table 3), with the median MFS in patients with baseline cT ≥ 2c being significantly shorter than in the remaining patients (92 months vs. not reached; 5 year MFS: 63% (95% CI: 45–76%) vs. 86% (95% CI: 72–93%); *p* = 0.003; Figure 1b).

A total of 57 out of 104 (55%) patients from the entire cohort survived until the end of the follow-up period, with a 5 year OS of 66% (95% CI: 56–74%; median OS: not reached; lower quartile OS: 39 months). Compared with the survivors, the non-survivors had significantly higher CTC counts, which were determined using the CellSearch^®^ system and more often yielded at least one CTC in this assay. Additionally, they presented significantly more often with PCa with a baseline cT ≥ 2. Finally, the group of non-survivors included all patients initially diagnosed with a metastatic PCa (Table 4).

The Kaplan-Meier curve for OS in the entire cohort is presented in Figure 2a. The analysis of univariate Cox proportional hazard models identified at least one CTC determined with the CellSearch^®^ system at the baseline (median OS: 28 months vs. not reached; 5 year OS: 30% (95% CI: 12–50%) vs. 76% (95% CI: 64–85%); *p* < 0.001; Figure 2b), a Gleason sum ≥ 8 (median OS: 66 months vs. not reached; 55% (95% CI: 36–71%) vs. 71% (95% CI: 59–80%); *p* = 0.044, Figure 2c), a cT ≥ 2c (median OS: 61 months vs. not reached; 5 year OS: 51% (95% CI: 36–64%) vs. 80% (95% CI: 66–88%); *p* = 0.009, Figure 2d) and metastases at initial diagnosis (median OS: 26 months vs. not reached; 5 year OS: not reached vs. 81% (95% CI: 71–88%); *p* < 0.001, Figure 2e) as significant predictors of a worse OS (Table 5). The presence of a metastatic PCa at the baseline turned out to be the only independent predictor of a worse OS when all those variables were analyzed together in a multivariate Cox proportional hazard model.

A total of 57 out of 85 (67%) patients without metastatic disease at the baseline survived until the end of the follow-up period, with a 5 year OS of 81% (95% CI: 71–88%; median OS: not reached; lower quartile OS: 63 months). Compared with the survivors, the non-survivors had significantly lower CTC counts, which were determined using the CellSearch^®^ system but more often yielded at least one CTC in this assay (Table 6).

The Kaplan–Meier curve for the OS in 85 patients with localized PCa at baseline is presented in Figure 3a. The analysis of univariate Cox proportional hazard models identified at least one CTC, which was determined with the CellSearch^®^ system at the baseline (median OS: 66 months vs. not reached; 5 year OS: 60% (95% CI: 25–83%) vs. 85% (95% CI: 73–92%); *p* = 0.014, Figure 3b) as the only significant predictor of a worse OS in that group (Table 7).

## 4. Discussion

This study, conducted in a group of 104 patients with high-risk PCa, demonstrated that a baseline CTC count ≥ 1, determined using the CellSearch^®^ system, was a significant predictor of a worse OS during at least a 5 year follow-up, also in the case of initially localized disease, but did not affect MFS.

While the fact that the baseline CTC count may determine a long-term prognosis in PCa in terms of the OS could be considered a clinically significant finding, the results of this study should be identified with caution. As previously mentioned, this study was an extension of our previous study, which analyzed the role of CTC count as a marker of disseminated disease in patients with newly diagnosed high-risk PCa [12]. In that study, patients with disseminated cancer presented with significantly higher CTC counts, determined with the CellSearch^®^, and identifying at least four CTCs with this system was the most accurate predictor of disseminated disease upon ROC analysis [12]. Importantly, in the present analysis involving the same group of patients, none of the subjects with disseminated disease survived five years after diagnosis. The presence of metastases at the baseline turned out to be the only independent predictor of a worse OS on multivariate analysis. Therefore, the presence of ≥ 1 CTC at the baseline should be interpreted primarily as a marker of existing PCa spread. Importantly, however, as shown in Table 2, a substantial proportion of the metastatic patients did not test positively for CTC, yet the spread had already occurred. Nevertheless, a CTC count ≥ 1 remained the only significant predictor of OS in a subset of 85 patients who initially presented with a localized PCa. The analysis of individual medical records demonstrated that 6 out of 10 patients with localized PCa and a baseline CTC ≥ 1 died during the follow-up period; this number included three patients without clinically detectable metastases who survived 25, 26 and 66 months from the CTC draw, respectively. Although the numbers are too low to formulate any firm conclusions, our observations might imply that a baseline PCa count determined with the CellSearch^®^ system might have a value as a determinant of long-term prognosis through the identification of patients with an occult cancer spread. Many previous studies demonstrated that the CTC number correlated with survival after treatment and could be applied as an accurate surrogate survival endpoint in patients with metastatic PCa [16,17,18,19,20,21,22,23,24,25,26,27,28,29,30]. In contrast, CTCs were rarely found in patients with localized PCa, which limited the prognostic value of this parameter [31,32,33]; the results of the present study seem to suggest otherwise.

The present study showed no link between the baseline CTC count determined with any method and the development of metastases in a subset of 85 patients who initially presented with localized PCa. This observation is not surprising given the long time that elapsed from the diagnosis of PCa to the detection of metastases in our patients. Judging from the lower quartile MFS of 63 months, it is highly unlikely that the metastases detected five years from the CTC draw and beyond were associated with tumor shedding at the time of diagnosis. This observation and the results published by other authors [34] imply that to be considered a prognostic marker in localized PCa, CTC counts should be monitored longitudinally at specific time intervals. Such an attitude would enable us to detect an occult, disseminated disease before it manifests clinically. In most studies mentioned above [16,17,18,19,20,21,22,23,24,25,26,27,28,29,30], including the one conducted by our group [12], CTCs were found at a time when metastases were already detectable using other techniques that are more routinely used in clinical practice.

Another issue identified in the present study is the reliability of CTC determination. We used three different methods, but the CTC counts determined with only one, the CellSearch^®^ system, were shown to have some prognostic value. In our previous study, the contingency between the CTC detection rates for the three methods was very low [12]. Discrepancies in the CTC numbers obtained with various methods were also reported by other authors [35,36,37]. In line with those findings, we found no link between the detection of CTCs, regardless of the assay used, and prognosis. CellSearch^®^ is the only method approved by the US Food and Drug Administration (FDA) to monitor breast, colorectal and prostate malignancies [38]. However, our observations and other published data suggest that the threshold of ≥ 5 CTCs per 7.5 mL, as recommended by the FDA, is reached by a relatively small proportion of patients and does not seem applicable in a clinical setting [7]. Further, the available evidence suggests that capturing an adequate number of cells for determining the prognostic or predictive value of CTCs would require the simultaneous use of more than one system [35] which, without a doubt, is not cost-effective. Hence, a foundation of future research on the application of CTCs in PCa should be optimizing a method for their enumeration.

The primary limitation of the present study stems from the lack of longitudinal CTC monitoring. Further, due to some technical problems, CTC counts determined with all three methods were not available for every patient; in particular, the number of patients with CTC numbers determined with the CellSearch^®^ system was relatively lower. Third, the survival analysis did not include treatment-related variables, as the main idea of the study was to verify whether the baseline characteristics could provide a long-term prognosis, and the group was too small to stratify it further according to the treatment variables. All these potential limitations should be considered when interpreting the results of the present study and planning future research.

## 5. Conclusions

The baseline CTC count can be considered a determinant of survival in high-risk PCa and in patients with localized disease. However, determining the prognostic value of the CTC count in patients with localized PCa would optimally require longitudinal monitoring of this parameter.

## Figures and Tables

**Figure 1 jpm-13-00608-f001:**
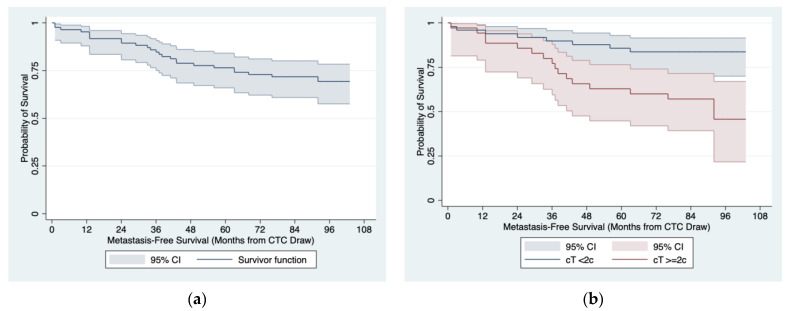
MFS estimates in 85 patients with localized PCa at baseline: (**a**) entire cohort; (**b**) by cT status at the baseline.

**Figure 2 jpm-13-00608-f002:**
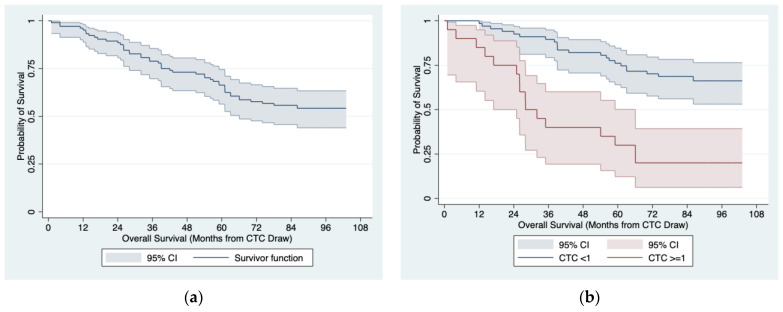
OS estimates in 104 high-risk PCa patients: (**a**) entire cohort; (**b**) by baseline CTC count ≥ 1 determined with the CellSearch^®^ system; (**c**) by baseline Gleason sum; (**d**) by baseline cT status; (**e**) by the presence of metastases at the baseline.

**Figure 3 jpm-13-00608-f003:**
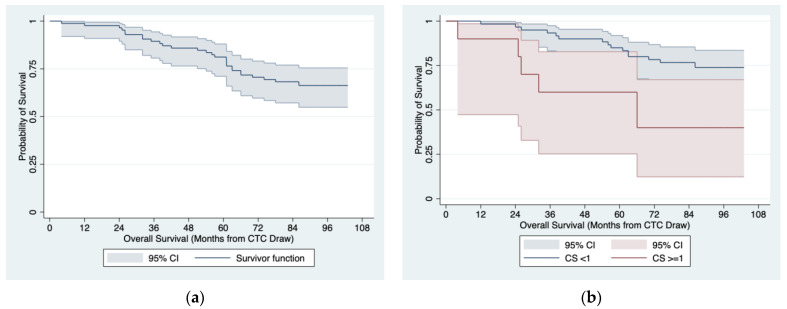
OS estimates in 85 patients with localized PCa at baseline: (**a**) entire cohort; (**b**) by baseline CTC count ≥ 1 determined with the CellSearch^®^ system.

**Table 1 jpm-13-00608-t001:** Clinical characteristics of the entire group of PCa patients participating in the study (*n* = 104) and the group of patients with localized PCa at baseline (*n* = 85), in addition to baseline CTC counts detected using the GILUPI CellCollector, EPISPOT assay and CellSearch^®^ system.

Variable	Entire Cohort (*n* = 104)	Localized PCa (*n* = 85)
PSA (ng/mL), median (range)	29.5 (0.5–191.0)	21.1 (0.5–191.0)
PSA > 20 ng/mL, *n* (%)	81/104 (78%)	66 (78%)
Gleason Sum (pts), median (range)	7 (6–9)	7 (6–9)
Gleason Sum ≥ 8 pts, *n* (%)	29/104 (28%)	19 (22%)
cT > 2c, *n* (%)	49/103 (48%)	35 (42%)
CTC (GILUPI), median (range)	1 (0–7)	1 (0–7)
CTC (GILUPI), *n* (%)	60/104 (58%)	50/85 (59%)
CTC (EPISPOT), median (range)	1 (0–25)	1 (0–25)
CTC (EPISPOT), *n* (%)	52/100 (52%)	45/82 (55%)
CTC (CellSearch^®^), median (range)	0 (0–569)	0 (0–54)
CTC (CellSearch^®^, *n* (%)	20/87 (23%)	10/70 (14%)
CTC (any assay), *n* (%)	90/104 (86%)	72/85 (85%)

**Table 2 jpm-13-00608-t002:** Baseline CTC counts detected using the GILUPI CellCollector, EPISPOT assay and CellSearch^®^ system and clinical characteristics of 85 patients with localized PCa at baseline, according to the development of metastases during the follow-up period.

Variable	Metastatic	Non-Metastatic	*p*
CTC (GILUPI), median (range)	1 (0–3)	1 (0–7)	0.493
CTC (GILUPI), *n* (%)	15/25 (60%)	35/60 (58%)	0.542
CTC (EPISPOT), median (range)	1 (0–25)	1 (0–4)	0.421
CTC (EPISPOT), *n* (%)	13/24 (54%)	32/58 (55%)	0.562
CTC (CellSearch^®^), median (range)	0 (0–3)	0 (0–54)	0.822
CTC (CellSearch^®^), *n* (%)	3/20 (15%)	7/50 (14%)	0.591
CTC (any assay), *n* (%)	21/25 (84%)	51/60 (85%)	0.572
PSA (ng/mL), median (range)	28.2 (2.5–191.0)	28.5 (0.5–136.9)	0.971
PSA ≥ 20 ng/mL, *n* (%)	17/25 (68%)	49/60 (82%)	0.138
Gleason Sum, median (range)	7 (6–9)	7 (6–9)	0.164
Gleason Sum ≥ 8, *n* (%)	7/25 (28%)	12/60 (20%)	0.296
cT ≥ 2c, *n* (%)	16/24 (67%)	19/60 (32%)	0.004

**Table 3 jpm-13-00608-t003:** Univariate analyses for MFS in 85 patients with localized PCa at baseline.

Variable	HR	95% CI	*p*
CTC (GILUPI)	1.04	0.47–2.34	0.907
CTC (EPISPOT)	0.96	0.43–2.13	0.913
CTC (CellSearch^®^)	1.13	0.33–3.88	0.838
CTC (any assay)	0.98	0.34–2.85	0.969
PSA ≥ 20 ng/mL	0.51	0.22–1.20	0.124
Gleason Sum ≥ 8	1.36	0.57–3.27	0.487
cT ≥ 2c	3.41	1.45–8.03	0.005

**Table 4 jpm-13-00608-t004:** Baseline CTC counts detected using the GILUPI CellCollector, EPISPOT assay and CellSearch^®^ system and clinical characteristics of PCa patients (*n* = 104), according to survival during the follow-up period.

Variable	Survivors	Non-Survivors	*p*
CTC (GILUPI), median (range)	1 (0–7)	1 (0–5)	0.071
CTC (GILUPI), *n* (%)	36/57 (63%)	24/47 (51%)	0.148
CTC (EPISPOT), median (range)	1 (0–4)	0 (0–25)	0.441
CTC (EPISPOT), *n* (%)	32/56 (57%)	20/44 (45%)	0.169
CTC (CellSearch^®^), median (range)	0 (0–54)	0 (0–61)	<0.001
CTC (CellSearch^®^), *n* (%)	1/49 (2%)	7/38 (18%)	0.011
CTC (any assay), *n* (%)	49/57 (86%)	41/47 (87%)	0.542
PSA (ng/mL), median (range)	29.3 (3.9–136.9)	30.0 (0.5–191.0)	0.900
PSA ≥ 20 ng/mL, *n* (%)	47/57 (82%)	34/47 (72%)	0.159
Gleason Sum, median (range)	7 (6–9)	7 (6–9)	0.064
Gleason Sum ≥ 8, *n* (%)	12/57 (21%)	17/47 (36%)	0.068
cT ≥ 2c, *n* (%)	21/57 (37%)	28/46 (61%)	0.013
Metastases at baseline, *n* (%)	0/57 (0%)	19/47 (40%)	<0.001

**Table 5 jpm-13-00608-t005:** Univariate and multivariate analyses for OS in 104 patients with high-risk PCa.

Variable	Univariate Model			Multivariate Model		
	HR	95% CI	*p*	HR	95% CI	*p*
CTC (GILUPI)	0.65	0.37–1.16	0.147			
CTC (EPISPOT)	0.68	0.37–1.23	0.199			
CTC (CellSearch^®^)	4.87	2.09–11.32	<0.001	1.24	0.48–3.20	0.654
CTC (any assay)	1.15	0.49–2.71	0.752			
PSA ≥ 20 ng/mL	0.67	0.35–1.28	0.226			
Gleason Sum ≥ 8	1.82	1.00–3.31	0.048	1.32	0.65–2.66	0.444
cT ≥ 2c	2.16	1.19–3.93	0.011	1.48	0.69–3.16	0.316
Metastases at baseline	14.41	7.08–29.30	<0.001	10.20	4.25–24.46	<0.001

**Table 6 jpm-13-00608-t006:** Baseline CTC counts detected using the GILUPI CellCollector, EPISPOT assay and CellSearch^®^ system and clinical characteristics of 85 patients with localized PCa at baseline, according to survival during the follow-up period.

Variable	Survivors	Non-Survivors	*p*
CTC (GILUPI), median (range)	1 (0–7)	0.5 (0–5)	0.085
CTC (GILUPI), *n* (%)	36/57 (63%)	14/28 (50%)	0.178
CTC (EPISPOT), median (range)	1 (0–4)	0.5 (0–25)	0.897
CTC (EPISPOT), *n* (%)	32/56 (57%)	13/26 (50%)	0.356
CTC (CellSearch^®^), median (range)	0 (0–54)	0 (0–3)	0.027
CTC (CellSearch^®^), *n* (%)	4/49 (8%)	6/21 (29%)	0.035
CTC (any assay), *n* (%)	49/57 (86%)	23/28 (82%)	0.435
PSA (ng/mL), median (range)	29.3 (3.9–136.9)	24.1 (0.5–191.0)	0.086
PSA ≥ 20 ng/mL, *n* (%)	47/57 (82%)	19/28 (68%)	0.108
Gleason Sum, median (range)	7 (6–9)	7 (6–9)	0.690
Gleason Sum ≥ 8, *n* (%)	12/57 (21%)	7/28 (25%)	0.440
cT ≥ 2c, *n* (%)	21/57 (37%)	14/27 (52%)	0.143

**Table 7 jpm-13-00608-t007:** Univariate analyses for OS in 85 patients with localized PCa at baseline.

Variable	HR	95% CI	*p*
CTC (GILUPI)	0.62	0.30–1.31	0.210
CTC (EPISPOT)	0.77	0.36–1.67	0.510
CTC (CellSearch^®^)	3.28	1.27–8.47	0.014
CTC (any assay)	0.81	0.31–2.13	0.665
PSA ≥ 20 ng/mL	0.54	0.24–1.19	0.128
Gleason Sum ≥ 8	1.17	0.50–2.76	0.716
cT ≥ 2c	1.74	0.81–3.71	0.154

## Data Availability

The datasets generated and/or analyzed during the current study are not publicly available due to patient privacy and the legal and administrative policies of the medical institution at which the study was conducted; however, they are available from the corresponding author upon reasonable request and approval from the Local Bioethics Committee.

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
