# Peer review of "Baseline CTC Count as a Predictor of Long-Term Outcomes in High-Risk Prostate Cancer"

_jpm, 2023, doi:10.3390/jpm13040608_

Round 1

Reviewer 1 Report

The authors have presented their findings in scientifically sound language with appropriate demonstration of results and tables. Methods are clearly stated and the whole manuscript is very well written. My recommendation for the current is manuscript is to be accepted with very mine corrections to the manuscript like check for the usage of prostate cancer after having introduced an abbreviation of PCa earlier.

Author Response

Thank you for your positive feedback. In line with your suggestion, we have abbreviated 'prostate cancer' as 'PCa' at the first use and used this abbreviation consistently throughout the text.

Reviewer 2 Report

Circulating tumos cells have been studied for several years as possible prognostic and predictive biomarkers for various tumors, including prostate cancer. In this interesting study, the authors evaluated the use of circulating tumos cells as predictors of 5-year overall survival and 5-year metastasis-free survival in patients with high-risk prostate cancer. The authors concluded that baseline circulating tumos cells count appears to be a predictor of overall survival but not metastasis-free survival, even in patients with localized prostate cancer.

Thank you to the authors for their contributions. The manuscript is well written, and the methods used are clear. The topic discussed is current and very interesting.

However, some minor publications are needed before publication.

 11)      As mentioned in the manuscript, this study has several limitations one of which is not having included in the survival analysis variables related to the treatment. However, it would be useful to mention in the manuscript the initial treatment to which the patients underwent. Were all patients treated with radiotherapy? If yes, with what treatment schedule? Was androgen deprivation therapy used in combination with radiotherapy? How were patients with metastases at diagnosis treated?

22)      In the multivariate analysis performed in this study, the presence of metastatic prostate cancer at baseline was found to be the only independent predictor of worse overall survival, whereas the other variables, including circulating tumos cells, did not prove to be independent predictors. How do the authors discuss this result?

Author Response

Thank you for your positive feedback. Please find below our responses to your comments.

1)      As mentioned in the manuscript, this study has several limitations one of which is not having included in the survival analysis variables related to the treatment. However, it would be useful to mention in the manuscript the initial treatment to which the patients underwent. Were all patients treated with radiotherapy? If yes, with what treatment schedule? Was androgen deprivation therapy used in combination with radiotherapy? How were patients with metastases at diagnosis treated?

Re: Following your advice, information about the treatment of the study patients has been added to the revised 'Treatment and follow-up' paragraph.

2)      In the multivariate analysis performed in this study, the presence of metastatic prostate cancer at baseline was found to be the only independent predictor of worse overall survival, whereas the other variables, including circulating tumos cells, did not prove to be independent predictors. How do the authors discuss this result?

Re: This issue has already been addressed in the second paragraph of the Discussion section. Briefly, the entire cohort included a subset of patients who presented with metastatic PCa at the time of enrollment, and all those patients died during the follow-up period. CTCs were not detected in a substantial proportion of those patients, yet PCa spread has already occurred. This explains why metastatic disease was a stronger (and the only independent) predictor of OS in the entire cohort. Nevertheless, CTC count ≥1 remained the only significant predictor of OS in a subset of 85 patients who initially presented with a localized PCa. The analysis of individual medical records demonstrated that 6 out of 10 patients with localized PCa and baseline CTC ≥1 died during the follow-up period.